# China's Forest Eco-Bank Project: An Analysis Based on the Actor-Network Theory

**Guangcheng Wei** [1,†]**, Xiangzhi Kong** [1,†]**, Yumeng Wang** [1,*,†] **and Qiang Gao** [2,†]

1 School of Agricultural Economics and Rural Development, Renmin University of China, Beijing 100872, China; wgc@ruc.edu.cn (G.W.); kongxz@ruc.edu.cn (X.K.)
2 College of Economics and Management, Nanjing Forestry University, Nanjing 210037, China; gaoqiang@njfu.edu.cn
* Correspondence: wymmyw@ruc.edu.cn; Tel.: +86-159-1085-7079
† These authors contributed equally to this work.

**Abstract:** The high degree of fragmentation and unsustainable exploitation patterns of forest resources have become prominent obstacles to the realisation of the economic and social value of China's forest resources. China's forest eco-bank (FEB) project was set up to achieve centralised utilisation of diffuse forest resources in an underdeveloped area. Analysing FEB projects is of great significance to countries with abundant forest resources aiming to benefit from the economic and ecological functions of such resources and achieve sustainable economic and social development. This study uses the actor-network theory framework to analyse cooperation among various actors in the implementation process of the FEB project in China. Our results indicate the following. First, the government was the principal actor playing a vital role in motivating the other parties and driving the project forward. Second, the diffuse nature of China's forest resources is a major obstacle to their utilisation. The centralised integration of forest resources is a prerequisite for their effective utilisation. Finally, it is necessary to improve the balance of interests of the various actors to promote a more stable actor network and attract more actors to participate in the FEB project.

**Keywords:** forest eco-bank; natural capital; forest resources; actor-network theory; ecological construction policy





## 1. Introduction

Global forest resources are a key means of tackling poverty [1–11] as they provide livelihoods for up to 1.6 billion people, including 250 million of the world's poorest [12–14]. Over the past 40 years or so, the Chinese government has implemented a series of forestry programmes to protect the ecological environment, which has resulted in a significant growth in forest area and timber stocks. China's forest cover has risen from 12.7% in 1970 to 22.96% in 2018 [15] and its forest area per capita is 0.15 hectares. Although it is the fastest growing forest resource in the world [16], China's forest cover—and its forest area per capita—is far below the world average [17].

Forest land fragmentation is a typical feature of China's current forest resources [18,19]. The reform of the collective forestry rights system in China is a major reform in the field of forest resources and has had a profound impact on the property rights of the country's forest resources today. According to data from China's State Forestry and Grassland Administration, the majority of China's forest resources are dominated by collective forest land; of the country's forest area, 84,366,100 hectares (38.66%) are state-owned, while 133,854,400 hectares (61.34%) are collective forest land. The decentralisation of forest land management subjects in China makes the problem of forest fragmentation more serious [19]. The Chinese government has made many explorations on the scale of forest resources, establishing 115,700 professional forestry cooperative organisations, 283,900 new business entities, carrying out forest rights mortgage loans and forest insurance, and

establishing 6602 forestry public service institutions nationwide. By the end of 2020, the area of contracted forest land management and the transfer for farmers' households reached 25,506.67 hectares, that of mortgaged forest land reached 3,780,400,000 hectares, and that of operating forest land reached 0.26 billion hectares, accounting for only 14.4% of the country's collective forest land area. However, the exploration of the large-scale use of collective forest resources is still very ineffective.

In 2009, the Chinese government put forward a daunting task for China's forestry industry by introducing a medium and long-term forestry development target to increase forest cover to over 26%, sustain stable ecological conditions, and promote a sound forest ecosystem and mature forest industry system by 2050 [20]. Although tree planting has been carried out to a large extent over the past few decades in China, it has not provided strong ecological benefits [21–23]. Many ecological restoration projects have merely pursued forest area growth through the afforestation or restoration of forest ecosystems. The relationship between ecosystems and the surrounding productive and living environment has not fully been analysed [24], the root causes of forest resource use have not yet been addressed [25], and conflicts between forest nature reserves and farmers' livelihood development still persist [26,27]. Thus, forest resources have not yet reached a truly restored state.

The relationship between population, economy, and natural resources is a difficult issue. Improving the income of rural residents in forest areas to help them achieve a stable escape from poverty is a challenge faced by the Chinese government. Since 1992, the income ratio between urban and rural residents has consistently remained above 2.5. Residents of forest-rich rural areas have only just lifted themselves out of poverty altogether by 2020 and are still at risk of returning to it. Despite establishing a strict system of forest nature reserves and supporting forest development through subsidies, the Chinese government has not yet been able to solve the problem of low income. To help address poverty in China and other developing countries, the economic value of forest resources must be realised so they can be used rationally [28,29].

Growing industrial output, economic growth, rising living standards, and structural changes in society have created higher demand for forest-related products, which has put more pressure on China's natural resources [30,31]. Forest-related service industries, such as tourism, recreation, and leisure are booming [32] as their development has significantly increased the economic contribution of forest resources [13,33]. In recent years, the goal of forest resource use has shifted from a self-sufficient, purely resource-exploiting approach to a diversified, commercial momentum [34]. Ecotourism is the fastest growing industry and has the most significant income-driving effect [35,36]. Even though diverse ways of realising forest values are emerging, the Chinese government still has not found a sustainable development path for the economic, social, and ecological value of its current model of forest development. It mainly offers protection through financial support. The 2018 Communiqué on the State of China's Ecological Environment states that China's financial expenditure on forest ecological compensation, returning farmland to forest and grass, grassland protection, and crop rotation alone amounted to 95.72 billion yuan in 2017. The type of financial subsidies is under enormous financial pressure, with a strong need for market-based compensation methods. The question of how to revitalise resources and integrate their value to achieve a win-win situation between economic development and ecological protection has long been a heated issue in the field of ecological economy and the environment [37].

Local governments in China have made a series of explorations and practices to unify the use of scattered forest resources. Among them, the Forest Eco-bank in Shunchang County is a typical practice [38]. In recent years, Shunchang County in Fujian Province is one of the cases that has achieved the sustainable development of forest resources, exploring the development path of building a forest ecological bank (FEB). By building an FEB platform, Shunchang County has promoted the economic value of forest resources while bringing into play their ecological value. Drawing on the 'decentralised input and centralised output (DICO)' model of commercial banks, Shunchang County relies

on the Forest Ecology Bank to centralise the collection and storage of scattered forest resources under its jurisdiction. They are then unified for management by professional operators, such as state-owned forest farms or tourism companies, realising the three-change reform of 'resources into assets, capital into shares and farmers into shareholders'. As one of the post-development areas rich in forest resources, Shunchang County carried out the innovative practice of forest eco-banking for green development in 2018, effectively exploring ecological construction in areas rich in ecological resources. This study aims to answer how China Forest Ecobank capitalised on forest resources and how the decentralised forest resources are integrated.

This study is developed through the following six parts: Section 1 introduces the current issues regarding forest resource utilisation in China. Section 2 provides a basic introduction to actor networks and analyses the reasons this theory is applied for the study of typical cases of FEB. Section 3 explains the reasons for selecting the case approach and the detailed process of data collection, and provides a basic introduction to the case. Section 4 follows the analytical framework of the actor-network theory (ANT) and provides an in-depth analysis of a typical case of FEB. Section 5 summarises the results achieved by the FEB in the past four years of operation. Section 6 draws the main conclusions from the case study section and presents corresponding policy recommendations.

## 2. Theoretical Analysis

ANT is used to study the relationships between stakeholders [39,40]. An actor network comprises multiple entities with related interests [41]. It provides a new perspective and method for studying the creation of networks of aligned interests [42]. ANT has greatly influenced the research on organisations and management [43–45] and is often used in research on sustainable development [46–49]. It is a useful theory for exploring the construction of an organisational network between heterogeneous subjects and 'opening the black box' to understand complex relationships between diverse actors [50]. We believe that ANT is appropriate for this study as it focuses on relationships between non-human and human entities, which is essential in the capitalisation of natural resources and the construction of organisations. ANT can be used to understand China's FEB, which is an innovative organisation led by the local government, with the participation of enterprises, rural households, and village collectives.

ANT has been used in tourism science [51–55], rural geography [56–60], and economic geography [58,60–62]. The theory has been continuously refined through research [59,63]. Given that ANT reveals the interactions between heterogeneous subjects, many scholars have used this feature to analyse the dynamic processes of rural resources and asset utilisation [55,56,64–66]. In China, where most of the country's forest resources are scattered in collective and individual hands, it is particularly important to centralise their use.

The global deforestation problem has received widespread attention [67], and many countries have adopted measures to address it [68]. The actor-centred power theory has become an important framework for analysing forest policy. It has been applied in a wide range of areas, including forest governance [68], participatory forestry [69], and policy formulation [70,71]. While there are other studies that have applied ANT to forest-related research [72–75], more extensive research is needed.

ANT comprises a relatively sound analysis framework, where everything in the social and natural worlds exists in constantly shifting networks of relationships. It can be divided into three main steps [76]. First, the human and non-human actors in the actor network must be determined, specifically those involved in the FEB. Human actors refer to individuals with subjective ability, including those who participate in various roles of FEB. Non-human actors mainly refer to subjects other than human beings, including all kinds of resources and elements, as illustrated in Figure 1. Second, the translation process of human and non-human actors, which primarily entails analysing the process of forming the FEB actor network, must be considered. Third, the effect of constructing the actor network must also be taken into account, primarily by analysing its construction. The translation

process is the most important step [77], as the position of actors in the network changes after translation [40].

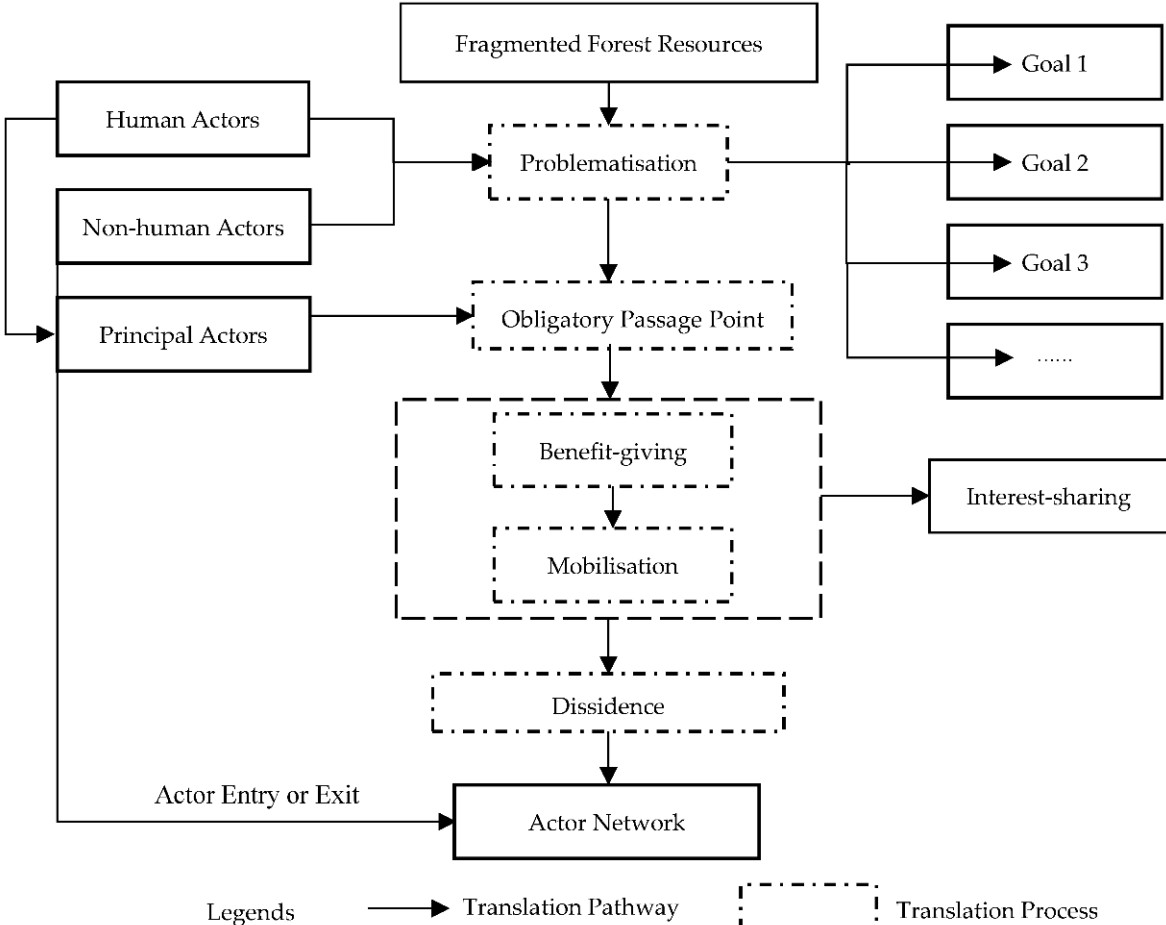

**Figure 1.** Research framework of the FEB actor network.

Rodger et al. [78] divided the translation process into five steps. The first step is 'problemisation', which means identifying the principal actor who first determined the specific path for actors to achieve their common interests and clarified the problems and demands of the human and non-human participants [79]. This prompts multiple actors to achieve a common goal [80]. Before the construction of the FEB actor network, different actors had various interest and demands. The Shunchang County government was the principal actor who first identified and then presented the demands and problems of the different actors. The second step is the 'obligatory passage point', which refers to the construction of the problem of the actor network in a certain way through new organisational forms and development measures so that heterogeneous actors can understand the value of the actor network. In the case of the FEB, the Shunchang County government was the principal actor who analysed and met the various demands of the actors through appropriate innovations to reach the obligatory passage point. The third step is 'benefit giving', the process by which the principal actor defines the roles of the other actors, assigns tasks, and ultimately shares benefits [81]. It requires finding ways to unite actors to construct the actor network. The fourth step is 'recruitment and mobilisation', which refers to the principal actor calling on the other actors to complete tasks, so they can be integrated into the action network and transformed into members. The fifth step is 'dissidence'. The network of actors is in constant change [41]. Actors need to be eliminated by continuous translation, due to differences in goals and interests. All actors have to involve in the dissidence stage. If not, it may lead to organizational disintegration [76].

The Shunchang County government used the FEB as a vehicle and platform to gather diffuse forest resources, centralise trading, and formulate various cooperation methods to attract forest workers, cooperatives, and village collectives to exchange or invest their natural resources. The fifth step is 'dissidence'. The actor network is constantly changing [41], and differences occur between actors due to varying goals or interests, which need to be dispelled through continuous translation [82]. The dissidence stage is important and involves all participants [50,82,83], as unresolved conflict may lead to the organisation disbanding [76]. In the case of the FEB, different actors may disagree with the actor network, and the principal actor will respond to problems faced by actors to solve difficulties and ensure the smooth operation of the organisation. These links will be applied to the case study discussed in this research. The actor-network translation process is shown in Figure 1.

## 3. Methods

### 3.1. Case Selection

We use the case study approach to conduct an empirical analysis for the following reasons. First, this approach involves the application of a typical case to analyse exploratory problems [84]. The research object of this study is China's FEB, which is an example of the Chinese government's successful exploratory practice of promoting sustainable ecological and economic development. Second, the case study method is suitable for studying events with several uncontrollable factors [85]. We look at the Shunchang County government as the principal actor who motivated the other parties and constructed the actor network and the factors it could not control, leading to its failure to meet the conditions of the control experiment. Shunchang County FEB officially started operating in December 2018 (The first "Forest Ecological Bank" is started in Shunchang County, Fujian Province. [DB/OL] Available online: http://grassland.china.com.cn/2019-01/09/content_40637891.html, accessed on 13 June 2022). The case of this FEB is typical, and we explore the reasons for the possible success of its construction of an actor network.

Our research team travelled to the city of Nanping in southeast China's Fujian Province to conduct field research in August 2020. The Shunchang County FEB was chosen as a typical case for the following reasons. First, in merely a few years since its establishment, it has created economic and ecological value from abundant local forest resources, which fits the research needs of this study. Second, Shunchang County motivated enterprises, different levels of governments, forestry centres, rural households, and other entities, such as guarantee companies and commercial banks, to participate in the construction of the FEB. Third, Shunchang County is a key forest area of Fujian Province, with abundant forest resources, making it a typical forestry county in China. Fourth, due to the low income of farmers, Shunchang County used to be a typical poor county in the province and has been identified by the provincial government as a key county for poverty alleviation and development [86]. In 2017, the county had 1553 poor households (4883 people), comprising 1156 households (3409 people) based on state criteria and 397 households (1474 people) based on provincial criteria, with 858 households (2389 people) eligible for subsistence allowances. Through the forest eco-banking project, local forest farmers have helped farmers in need increase their income and escape poverty by revitalising their forest resources. The locations of the study areas are illustrated in Figure 2.

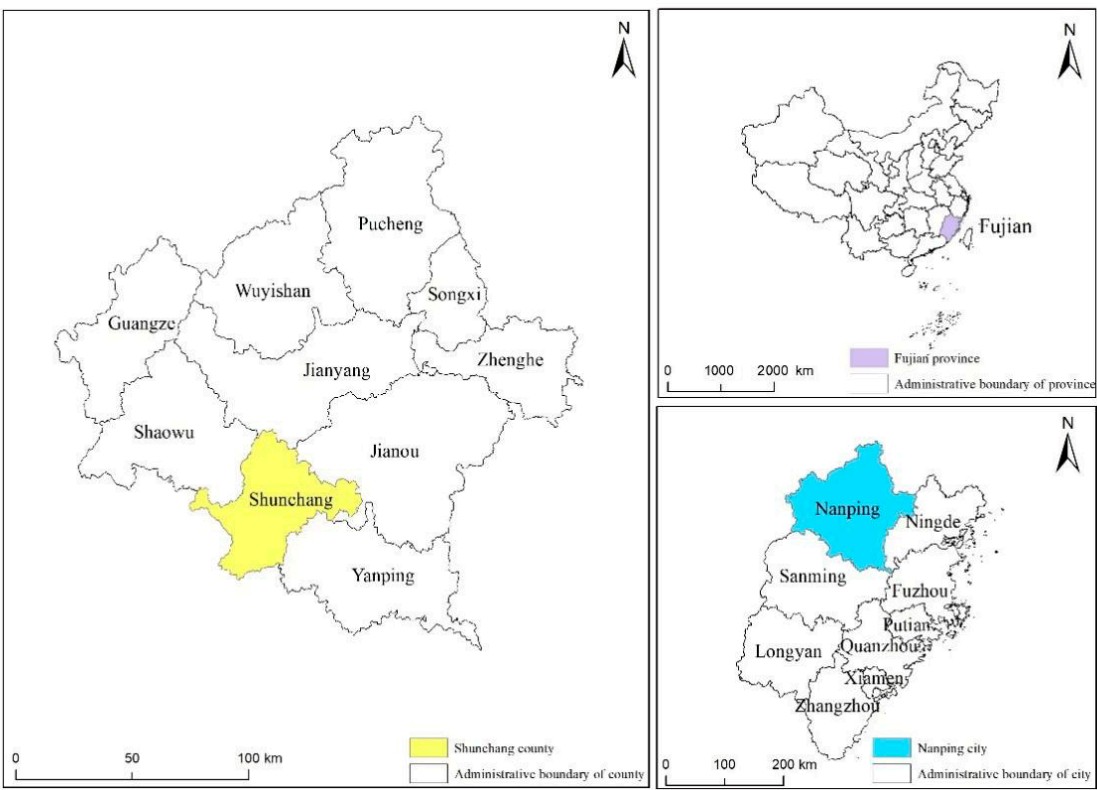

**Figure 2.** The location of the study areas.

Shunchang County is a National Model Green Unit and the forest county and garden city of Fujian Province. It is rich in forest resources, with 80.5% [38] forest cover and a forest stock volume of 15.64 million m$^3$ [38]. Its area of forestry land in is 166,000 hectares, including 39,300 hectares of state-owned forest farms and 126,700 hectares of collective and individual forest land [87]. Despite 76% of Shunchang County being covered by forest, its forest land is fragmented and diffuse in rural households [88]. Determining how to manage such a large volume of fragmented forest resources was the priority of the Shunchang County FEB. To address these issues as well as the high costs and low incomes of single-family operations, Fujian Province established the FEB operating platform for the Shunchang County State-Owned Forest Farm. By transforming, integrating, protecting, and promoting the diffuse and fragmented forest resources, it realised diversified value addition to forest resources, supported the transformation of forest resources into funds and assets, and promoted the integrated development of primary, secondary, and tertiary forestry industries.

In accordance with the principle of ensuring government direction, through rural residents' participation, market-based operations, and the primacy of enterprises, the Shunchang County government established an FEB that harnessed the strengths of various participants to oversee the utilisation of forest resources. Lvchang Forestry Resources Operation Company registered a capital of CNY 30 million and is the main market-oriented entity of Shunchang County's FEB [89]. Shunchang County's state-owned forest farm has a controlling stake in the company and eight community-level state-owned forest farms have shares in it. The company consists of two centres, which oversee data management and asset evaluation and purchasing, and three companies, which are in charge of forest operations, trusteeship, and financial services. The former provides data and technical support, whereas the latter is responsible for purchasing, stocking, entrusting, operating, and upgrading resources. The FEB also brings together the resource station of the county forestry bureau, the state-owned forest farm's felling area survey and design team, and community forestry and protection teams to carry out resource management and pro-

tection, resource assessment, transformation and upgrading, project design, operations development, and forest tenure changes. The FEB encourages forest farmers to voluntarily transfer fragmented forest resource management rights and use rights to the FEB without the need to change the ownership of the forest land. The FEB aims to implement centralised reserves and scaled-up management via a rational approach, intensive management, and development of an under-forest economy to achieve clear ownership and centralised and contiguous high-quality collection of assets.

*3.2. Case Data*

In accordance with the principle of triangulation, our research team obtained data through multiple channels to ensure reliability. The case data collected in this study came mainly from three sources. First, semi-structured, in-depth, one-to-one interviews lasting 1–2 h were conducted with people in charge at the county, town, and district levels: village officials; enterprises; banks; and rural residents involved in the construction of the FEB. Investigators also encourage other members to supplement and participate in discussions on certain issues. A total of 89 questionnaires were collected in this survey, including 9 government employees, 6 eco-bank employees, 11 enterprise managers, 5 cooperative directors, 12 village leaders, and 46 farmers. Our research team transcribed and encoded the interviews. The word count of the transcripts ranged from 5000 to 15,000, totalling 60,000 words. Second, we conducted site visits of local FEB cases. Finally, we collected other data, including information on the construction of the FEB through our research team's discussions with the Nanping Municipal Agriculture and Rural Affairs Bureau, county and municipal governments, enterprises, and rural residents, to gain a comprehensive understanding of the processes, problems, innovations, and extensive relevant publicly available information, including documents and reports on China National Knowledge Internet, WeChat official accounts, and Baidu (These well-known Internet sites can represent most online information sources in China).

## 4. Actor-Network Analysis of FEB

### 4.1. Composition of Actors

ANT mainly focuses on interconnections between human and non-human actors who interact in space. Therefore, according to the research of Callon (1984) [76], this study divided the actor network of the FEB into two parts: human actors and non-human actors. The former includes the Shunchang County government, Lvchang Forestry Resources Operation Company, Lvchang Forestry Financing Bonding Company, forest farmers, forestry cooperatives, village collectives, the FEB, financial institutions, bonding companies, state-owned forest farms, and processing enterprises. Non-human actors include forest resources, industrial and commercial capital, and financial loans. The specific composition is given in Table 1.

**Table 1.** Composition of the FEB actor network.

| Type | Category | Actor |
|---|---|---|
| Human actors | Principal operator | FEB, Lvchang Forestry Resources Operation Company, Lvchang Forestry Financing Bonding Company |
| | Non-principal operator | Forest farmers, forestry cooperatives, village collectives, tourism companies, financial institutions, bonding companies, state-owned forest farms, processing enterprises |
| | Government | Shunchang County government, all levels of town government |
| Non-human actors | Core resource | Forest resources |
| | Non-core resource | Capital, financial loans, bonds |
| | Other factors of production | Land, labour |
| | Carrier | Forest landscape, large-scale forest farms |

Source: From research data.

*4.2. Obligatory Passage Point*

　　Before launching the FEB pilot project, heterogeneous actors in Shunchang County had their own interests, demands, problems, and obstacles (Figure 3). In terms of the human actors, the Shunchang County government was committed to using abundant local forest resources to their full potential, achieving poverty alleviation goals, increasing rural residents' incomes, developing the local forestry economy, and realising ecological sustainable development. Local financial institutions responded enthusiastically to the national call to participate in ecological construction, but the forestry industry lacked collateral to protect financial institutions from risks. As a state-owned enterprise, the Nanping Rongqiao Bonding Company wanted to fulfil its social responsibility by participating in an ecological industry, but forestry resources were relatively diffuse and difficult to concentrate. As the forestry industry has traditionally achieved low profitability, the company struggled to find high-quality projects that it could guarantee. Of the forest land in Shunchang County, 76% was in the hands of collectives and individuals, and only 24% was managed by state-owned forests [88]; thus, the state-owned company was limited by its small scale and low efficiency. State-owned forest farms had tried various ways to manage the forest land of collectives and individual forest farmers, but these efforts were halted due to the diffuse nature of forest land and difficulty conducting large-scale operations.

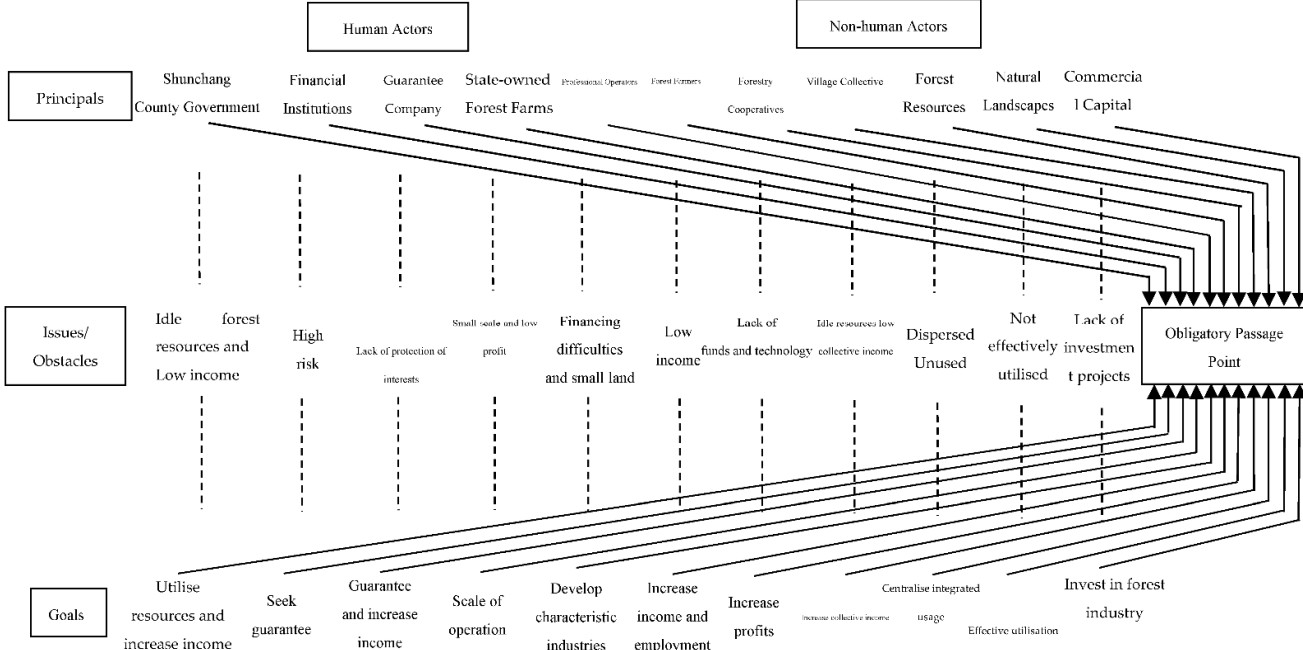

**Figure 3.** The interest requirements and objectives of the various actors of the FEB.

　　Village collectives wanted to develop and expand the collective economy by revitalising collectives and subcontracting forest land to households. Poor villagers hoped to effectively use this subcontracted forest land to increase their property income and extricate themselves from poverty as quickly as possible. However, the number of residents who rely on forestry for their livelihoods has been decreasing significantly. Poor rural residents found it difficult to find jobs, especially locally. Many ordinary residents in Shunchang County work in cities away from home, which has continuously increased the number of part-time employees. In terms of forest land, a large area is not being used effectively. Professional operators had difficulties finding large areas of contiguous forest to develop and faced financing challenges, which hindered the creation of new business formats in rural industries. Forestry cooperatives were limited by their small scale, rendering them unable to take advantage of economies of scale. Additionally, they had varying production standards, which did not always meet the needs of leading companies.

Regarding non-human actors, the area's abundant forest resources were also not being used effectively. Most forest land rights belonged to households, so resources were fragmented and difficult to aggregate or use on a large scale. The forest resources of Shunchang County were still in the exploratory stage of their product life cycle. Product types were not diverse, and profits were mainly from logging. There was a lack of sustainable development capabilities and an urgent need for industrial upgrading. Moreover, it was difficult to efficiently utilise high-quality forest resources in Shunchang County. Local natural resources had not been adequately evaluated and priced, making it impossible to quantify their true value or know the ecological service functions of the natural capital. This hindered the marketisation of forest resources. Furthermore, in terms of utilising forest resources with industrial and commercial capital, there was conflict between the 'big market' and the 'small rural residents'. The level of market-based operations involving forest resources was low, and it was difficult for industrial capital to gain a foothold. There were also issues concerning communication with rights holders and excessive transaction costs during resource exploitation activities by outside entities. Such problems force actors to find points of convergence and ways to realise their interests and goals. The FEB was a way to integrate and utilise local forest resources and motivate various actors to participate.

In case of the human actors, the Shunchang County government utilised local forest resources, realised their economic function while ensuring their ecological function, increased rural residents' incomes, and completed its poverty alleviation work. Local financial institutions and the Nanping Rongqiao Bonding Company provided loans and guarantees for projects with acceptable risk, supported and guaranteed by the government. Thanks to the FEB, the state-owned forest farm realised the large-scale management of diffuse forest resources and economies of scale. Village collectives made full use of their collective forest resources and increased collective income. By cooperating with the state-owned forest farm, cooperatives reduced operating costs, improved the quality of forest products, and met the high standards of leading enterprises. Ordinary rural households handed over their idle forest resources to the FEB for unified contracting and management, which increased their property income. Poor households transferred their forest land to the FEB to increase their incomes and achieve stable poverty alleviation by stimulating employment.

As for the non-human actors, forest resources were used sustainably with the FEB system through the centralised and large-scale utilisation of diffuse forestry resources.

The FEB system facilitated the flow of industrial and commercial capital, centralised the development of forest resources, and encouraged the participation of actors in overcoming high transaction costs to revitalise those resources. Loans and guarantees were combined to provide financial support for projects that utilise forest resources. The FEB organised the effective planning and utilisation of land, thereby overcoming previous difficulties of being diffuse and difficult to utilise. Previously, it was difficult to utilise idle labour, but this changed with the development of new rural industries and forms of business and large-scale forestation. Due to diffuse resources and a lack of planning, new industries and forms of businesses were difficult to promote but were realised thanks to the 'centralised output' function of the FEB. The large-scale forest farm was previously operated by a state-owned company, but it accounted for only 23.63% (According to the data of Shunchang Statistical Yearbook) of the total county's forest area and needed to expand. The county's total forest land was already subcontracted to households, so forest resources were diffuse and could not be developed and utilised in a centralised manner. Subsequently, a large-scale forest farm was established using the 'centralised output' function of the FEB.

As the core actor, the Shunchang county government organised and led other actors to build the core organisation of the FEB with human and non-human actors. The FEB contributed to sustainable ecological development using local land and labour resources after acquiring 4170 hectares of forest land and expanding the management scale of state-owned forest farms by 9.78% [90]. The FEB promoted regional economic development by implementing carbon sequestration projects, which boosted the income of rural residents by over CNY 1 million. It helped alleviate poverty in the country by increasing the

income of 154 poor households through employment. It is now easier for industrial and commercial capital to reach the countryside, with investments of over CNY 1 million in seven projects under construction with 14 having completed the planning stage [91]. New rural industries and business formats have been developed, including more than 20 leisure and entertainment venues, such as innovation parks, cultural villages, and amusement parks [88]. The FEB project helped achieve the objectives of both human and non-human actors, becoming the obligatory passage point in the formation of the actor network.

### 4.3. Benefit-Giving

In the empowering stakeholders stage, the principal actor motivates the other actors through benefit-sharing, clarifies their roles, and promotes the analysis of problems and the accomplishment of objectives. To ensure that actors with aligned interests continue to play a role in the actor network, it is necessary to ensure balanced and stable relationships between them. Benefit-giving is a measure used by the principal actor to stabilise relationships among actors with aligned interests. It can encourage actors to play the new roles assigned to them by the principal actor and achieve a balance between heterogeneous actors. In the case of the FEB, the different actors have varying interests. When their interests diverge, conflicts may occur. To avoid conflicts of interest destabilising the FEB's actor network, it is necessary to go through the obligatory passage point in the interest stage so that the various actors form an aligned network around the obligatory passage point and their own interests. The interest requirements and objectives of the various actors of the FEB are shown in Figure 4. The new roles of the actors are defined through policies, resource allocations, and capital investment, which motivate them to play active roles. We summarized the process of identity transformation of multiple actors as Figure 4, according to the research of Wang and Zhao (2021) [92].

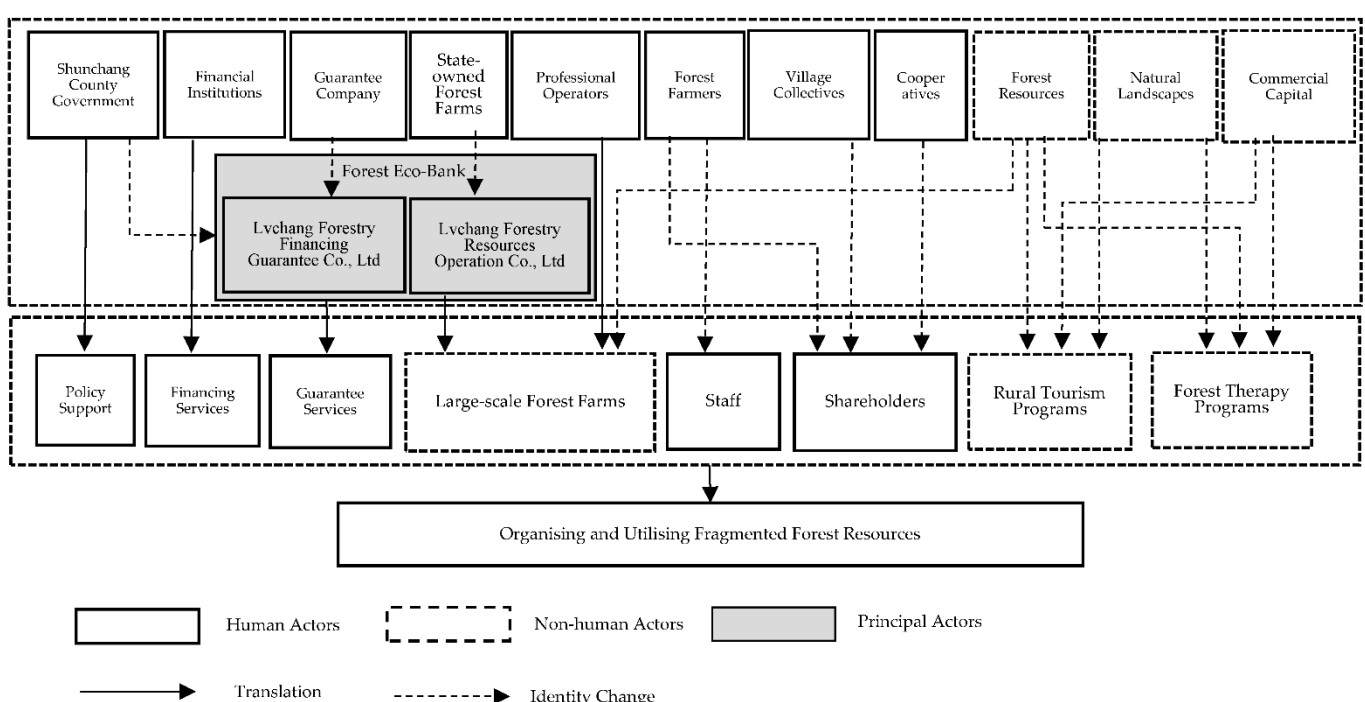

**Figure 4.** The FEB actors' network and their mechanisms.

To motivate heterogeneous actors, the principal actor must be able to analyse the problems and demands of the actors, find an obligatory passage point, and grant benefits. The Shunchang County government—the principal actor—was interested in utilising forest resources, developing the regional economy, helping rural residents increase their incomes, and alleviating poverty. To encourage the various entities to participate, the

county government organised eight township forest farms for the Shunchang County State-Owned Forest Farm and for the Nanping Rongqiao Bonding Company. This was done to establish the Lvchang Forestry Resources Operation Company, and the Lvchang Forestry Financing Bonding Company as the operating companies of the FEB, provide centralised operations and financing services for forest resources, and solve obstacles facing the development of the industry. Lvchang Forestry Resources Operation Company consists of the Data Management Center, Asset Evaluation and Purchasing Center, Lvchang Forest Management Company, Lvchang Trusteeship Company, and Forestry Financial Services Company. The Data Management Center is responsible for compiling information about stocks of forest resources and building a resource ledger, as well as keeping them up to date. A strict system that clarifies property rights helps when seeking to capitalise the value of natural resources to address the low utilisation of forest resources by the Shunchang County government. The Asset Evaluation and Purchasing Center evaluates the prices of natural resources that need to be exchanged to protect the legitimate rights and interests of all parties. This is to address the financing risks of financial institutions, bridge financing companies, and protect the vital interests of smallholder farmers. The three subsidiary companies of the Lvchang Forestry Resources Operation Company are responsible for the purchase, trusteeship, operation, and collateral financing of resources. Lvchang Forest Management Company and Lvchang Trusteeship Company are responsible for the transfer of rights and trusteeship of forest land. The former has used state-owned forest farms to overcome the problems of forest farmers' operations being small, chaotic, and diffuse; meet the high standards of leading enterprises; and increase forest farmers' income through a dividend system. The latter replaced cooperatives, village collectives, and forest farmers as the operator of forest resources. By charging a trusteeship fee, the Lvchang Trusteeship Company helped improve efficiency and increase rural residents' incomes by introducing high-quality products and professional services. Lvchang Forestry Financing Bonding Company issues credit and financing to business entities based on evaluation reports by the data and asset centres to overcome the shortage of enterprise funds. The company was jointly established by the Shunchang County government and the Nanping Rongqiao Bonding Company. It is responsible for meeting the financing needs of professional operators; providing forestry rights collateral guarantee services to forestry enterprises, collectives, or forest farmers who require financing; and managing loans collateralised by forest rights guaranteed by the FEB. Such loans have a term of up to 15 years and annual interest of 5.74%, which is nearly 50% lower than an ordinary bank loan [93]. Market-based financing and professional operations have solved the capital needs for transferring rights and stockpiling [94]. Village collectives, cooperatives, and rural households have increased their incomes by participating in the FEB project. Poor households have managed to shake off poverty by transferring forest land rights and gaining employment. Professional operators have realised the large-scale transfers of forest rights through links with the FEB, solved financing difficulties, and promoted rural industries and new forms of business.

The Shunchang County government was instrumental in the development of the FEB at this stage. It played an important role as the key actor in granting benefits to actors through organisation, leadership, publicity, and demonstration. The government solved the heterogeneity problem of diversified subjects, unified their goals, formed a cohesive force, and built a close network of actors through shareholding and circulation. It developed an organisational structure for the FEB to provide professional operators with financial support such as loans, which expanded channels for forest farmers to increase their incomes. It also created jobs, ensured the risks of financial institutions and guarantee companies, and realised the large-scale utilisation of ecological resources.

*4.4. Recruitment and Mobilisation*

Through recruitment and mobilisation, the majority of heterogenous participants can be encouraged to participate in the development of the collective economy. The

orderly participation of farmers can be encouraged and their fundamental interests realised. Mobilisation is closely related to the interests of actors, and its premise is interest correlation. After mobilisation, the construction of the actor network is completed. Actor networks need to mobilise both human and non-human actors. The role of core actors is to endow heterogeneous actors with tasks and interests and make them become members of an interest alliance. In the mobilisation stage, the principal actor represents all the entities in the network and motivates other members to act around the obligatory passage point.

In the stage of recruitment and mobilisation, typical cases still have strong common problems and characteristics. First, forestry farmers are often risk averse when pursuing stable income or minimum dividends. They are unwilling to transfer forest land accessories to the FEB at one time. Second, financial institutions, guarantee companies, professional operators, and other investors are usually more cautious to prevent risk. Third, professional operators such as state-owned forest farms and tourism companies tend to operate large-scale forest land rather than fragmented land. Since most of the forest land in Shunchang County is held by rural collectives and farmers, it is difficult to make centralised use of it due to the phenomenon of resource fragmentation, which has perplexed state-owned forest farms and professional operators for a long time. Fourth, the Shunchang County government as the core actor, focuses on protecting the rights and interests of farmers and investors during the recruitment and mobilisation stage, which is the key to realising recruitment and mobilisation. Fifth, the neighbourhoods' social network in rural society is relatively close. By encouraging a small number of farmers to participate in the FEB system, mobilisation can be realised.

In the initial stage of the FEB, the concerns of heterogeneous actors lead to their low enthusiasm to participate in the actor network, which needs to be mobilised. In the process, Shunchang County government, as the core actor, targeted specific problems given the differentiated confusion and doubts of heterogeneous actors, fully mobilised the participation enthusiasm of diversified subjects, and successfully realised mobilisation, which is illustrated in Table 2.

**Table 2.** Issues encountered in the recruitment and mobilisation of typical cases and their solutions.

| Issues | | Solutions | |
|---|---|---|---|
| **Actors** | **Details** | **Actors** | **Details** |
| Forestry farmers | Pursuing steady returns | FEB | Multiple transfer modes |
| Rural collectives | | Shunchang County government | Demonstration |
| Commercial banks | Lack of collateral | Guarantee companies | Provide guarantee |
| | Difficult to measure asset value | FEB | Valuation of assets |
| Guarantee companies | Lack of risk guarantee | Shunchang County Government | 12 million yuan contribution |
| State-owned forest farms | Lack of scale forest farms | FEB | Centralised and continuous revitalisation |
| | High investment risk | FEB | Guarantee long-term use right |
| Tourism companies | Lack of large-scale landscape | FEB | Centralised and continuous revitalisation |

### 4.4.1. Forestry Farmers and County Collectives

The FEB's initial plan to coordinate the use of forest land through centralised stockpiling by village committees did not work as planned. Due to a lack of awareness and channels to access information, there was confusion and suspicion among forest farmers, village collectives, and cooperatives about the role of the FEB. Consequently, they were unwilling to give up the work and lifestyle to which they were accustomed and to transfer

their land for the FEB's centralised use. Even older rural residents assumed it was a form of disguised repossession and refused to transfer their land.

As a result, the Shunchang County government arranged for township governments and village committees to conduct mobilisation work. Village committees transferred forest land not subcontracted to households to the FEB and encouraged village collectives to invest land for poor households as an example to others. For example, the first poor household to join the FEB project deposited its 0.6 hectares of young Chinese fir forest to the FEB at the end of 2018. According to the terms of the agreement, the household would receive monthly payments of the anticipated profit amounting to CNY 310 from the FEB for the next 20 years. Upon expiration of the trusteeship period, the household will receive 60% of the income from the sale of timber, depending on its price, after deducting management and protection costs. After three months of mobilisation work, by the end of February 2019, the FEB had completed the buyout of 22 households with a total area of 296 hectares, signed two trusteeships with a total area of 0.9 hectares, processed 11 forest rights collateral loans worth a total of CNY 4.85 million, and mortgaged forest land covering a total area of 131 hectares. Eleven towns and villages have already transferred 100% of their forest land to the project. Thanks to demonstrations and guidance, increasingly more rural households, especially those with people who travel to cities for work, have joined the FEB project.

Although the FEB has gained the trust of most rural households, cooperatives, and village collectives, the demands of the various actors differ depending on their circumstances, and some actors still harbour doubts about participating in the FEB project. To ensure the individual needs of forest farmers, village collectives, and forestry cooperatives are met based on their specific circumstances, the FEB offers the four transfer methods of stock ownership, trusteeship, lease, and buyout. Under stock ownership, forest farmers who wish to be involved in joint operations buy shares with the contract rights to one logging cycle and the forest assets. The profit is split between the forest farmer and the FEB usually in a 3:7 or 4:6 ratio, and the state-owned forest farm determines how it is managed. Thus, forest farmers become shareholders and gain a share of the income. The main subscribers to the trusteeship method are poor households who entrust their forest land and trees to the FEB after the trees are logged. During the trusteeship, the subscribers benefit from a fixed dividend each year, and receive more than 60% of the profit after felling. Under the leasing method, if forest farmers have idle forest land, they can lease it for a logging cycle in return for a rental payment. During the logging cycle, the FEB pays the lessor according to the annual lease value of CNY 900 per hectare for Type I and Type II land, and CNY 600 for Type III. When felling is completed, dividends are also given to forest farmers based on land area. In the buyout method, if forest farmers no longer wish to manage their land, the FEB will outright buy their forest land management rights and forest ownership at a fair market price. Those who wish to realise their assets can transfer their forest ownership and contracted land-use rights to the FEB in accordance with the requirements of the Implementation Plan of Shunchang County for Commodity Forest Redemption. By the end of June 2019, the FEB had completed buyouts of 2133 hectares of commercial forests, leased 773 hectares, and contributed to the afforestation of 68 hectares, including full coverage of the administrative villages of Shunchang County.

### 4.4.2. Commercial Banks and Guarantee Companies

Although banks have the task requirements of serving agriculture, rural areas, and farmers, they are still cautious about issuing unsecured loans due to the lack of collateral. In order to solve this issue, Shunchang County government introduced a joint stock with the Nanping Rongqiao guarantee company to establish 'Fujian Shunchang Lvchang forestry financing guarantee company' and provide guarantee loan services for state-owned forest farms and forest farmers. Although Lvchang guarantee company was jointly established by the government and the Rongqiao company, the Rongqiao guarantee company was concerned about the risk. In this regard, Lvchang county government had many discussions with the Rongqiao company before finally deciding to jointly establish a guarantee company

by investing CNY 12 million, and bear the risk according to the investment ratio of 6:4, which has solved the worries of the Rongqiao company.

### 4.4.3. State-Owned Forest Farms and Tourism Companies

As the main body of large-scale management, state-owned forest farms tend to operate concentrated and contiguous woodlands. On the one hand, in the early stage of the construction of the FEB, the state-owned forest farm was worried that the forest land transferred by the FEB was scattered and fragmented, which would make giving full play to the economies of scale difficult. Moreover, the state-owned forest farm was unwilling to cooperate with the FEB. In order to solve this problem, Shunchang County government and the state-owned forest farm have determined the lower limit of the operation scale of 3.33 hectares. Forest land below 3.33 hectares is managed by cooperatives, large-scale forest farmers, and other entities, and forest land of 3.33 hectares and above is uniformly managed by the state-owned forest farm. On the other hand, forestry is an industry with a long investment cycle. In the short term, it is mainly investment, which is difficult to ensure the stability of land management rights. Therefore, because of many investigations and visits, Shunchang County government decided to set the circulation period above at least one rotation period, supplemented by various circulation modes such as shareholding, trusteeship, leasing, and buyout.

As an important carrier for the development of new forms of rural industry, tourism companies tend to forest land with beautiful landscapes, stable management rights, and concentrated operations. On the one hand, when tourism companies develop areas with beautiful natural landscapes, the asset specificity is high, but the land management right is often unstable, which makes tourism companies reluctant to invest. To solve this issue, the FEB fully guaranteed the operation right of tourism companies in the form of contracts, set the circulation operation period at more than 10 years, and required that tourism companies can give priority to renewal to ensure the long-term use right of tourism companies. On the other hand, due to the serious fragmentation of forest resources, it is difficult for tourism companies to find large-scale forest landscape.

### 4.5. Dissidence

Empowering stakeholders is important for the stability of the interest alliance. The dissent of heterogeneous actors on the actor network in the interest alliance helps promote the renewal of the actor network and improve the coordination of all parties' interests. In the objection link, full consultation and communication is a necessary condition to ensure the long-term stability of the interest alliance relationship. In the objection stage, the problems and solutions of typical cases still have strong commonalities, which are embodied in the following three aspects.

First, small-scale business entities are unable to give full play to the advantages of economies of scale. Second, the main provider of factors is still an actor prone to dissent. Commercial banks often pursue stable income and low risk, whereas tourism companies provide more guarantee for loans and investment to reduce guarantee risk. Third, the FEB and Shunchang County government can play an important role in resolving objections, and adjust the distribution plan through communication and consultation to meet diversified demands. After the recruitment and mobilisation of typical cases, the objections of heterogeneous actors on interest distribution and risk guarantee were handled according to local conditions, and the actor network was updated and optimised, as shown in Table 3.

**Table 3.** Issues encountered in the dissidence stage and solutions of typical cases.

| Issues | | Solutions | |
|---|---|---|---|
| **Entities** | **Details** | **Entities** | **Details** |
| Small-scale businesses entities | Low efficiency | FEB | Uniformly entrust state-owned forest farms to operate |
| | Low quality | Shunchang county government | Cooperation with scientific research institutes and forest enterprises |
| Guarantee companies | Lack of guarantee | Shunchang county government | Split risk in the proportion of 8:2 |
| State-owned forest farms | Capital shortage | Shunchang county government | Promote the listing of state-owned forest farms |
| | | FEB | Introduce new varieties and develop understory economy |
| | | Commercial bank | Offer low-interest loans |
| Tourism companies | Capital shortage | Guarantee companies | Tender guarantee |
| | | Commercial bank | Offer low-interest loans |

### 4.5.1. Small-Scale Business Entities

Regarding the extensive management of small-scale business entities, the FEB initially only handed over large areas of forest land to the state-owned forest farms for management. It gave management contracts of small areas of forest land to professional households or cooperatives. Eventually, it became apparent that although professional households and cooperatives had certain scale and technical advantages compared to individual farmers, they suffered from extensive management and low efficiency. To solve this issue, the FEB decided to use the state-owned forest farm as its dedicated management unit. Furthermore, forestry operators in Shunchang County had the problem of substandard product quality. Shunchang County implemented the National Reserve Forest Quality Improvement Project and carried out directional technology introduction with Fujian Academy of Forestry Sciences and Shengsheng Wood Industry Enterprises. It adopted four changes: it moved from clearcutting to selection-cutting, from single-storied stands to multi-storied stands, from only coniferous trees to a mix of coniferous and broad-leaved trees, and from general timber to special native rare timber. The FEB optimised the forest stand structure, increased forest stock, improved the quality and value of forest resources, gained Forest Stewardship Council International Certification, standardised the management of traditional forest areas, and provided support for exporting processed timber products to European and American markets.

### 4.5.2. Guarantee Companies

Although Shunchang County government has invested CNY 12 million and agreed to bear the corresponding risks with the Rongqiao guarantee company in the proportion of 6:4, the Rongqiao guarantee company found that the real risk in the actual operation is much higher than expected. Moreover, the Rongqiao guarantee company believes the government should split the risk. After many exchanges, Shunchang County government and the Rongqiao guarantee company finally agreed that both parties should bear the risk of bank bad debts in the proportion of 8:2 to solve the dissidence of the guarantee company.

### 4.5.3. State-Owned Forest Farms and Tourism Companies

The shortage of funds faced by state-owned forest farms has gradually become prominent. According to the contract, the operating company needs to regularly distribute income to the transferor, which has great capital pressure. In addition, the forest ecological bank attracts more and more farmers, which gradually makes the operation of state-owned forest farms unsustainable. The Shunchang County government and the FEB attempted to

solve the existing problems in state-owned forest farms from the following aspects. First, it began to promote the listing of state-owned forest farms, start the listing of Shunchang County State-owned Forest Farm in 2019, and explore the development road of socialised financing. It was in May 2020 that the signing with law firms, accounting firms, and securities companies was completed, and the mobilisation investigation was carried out to complete the enterprise restructuring as soon as possible and be listed within 3–5 years. Second, FEB innovated and built financing projects. Under the overall planning of Nanping City, the Shunchang County government planned to implement the public-private partnership project of the first national reserve forest precision improvement project in China, and obtained a credit of CNY 912 million from the China Development Bank (CDB). This effectively accelerated the intensive plantation cultivation, forest rights purchase, and forest transformation and cultivation projects of the 'forest ecological bank', providing sufficient funds for the large-scale operation of state-owned forest farms. Finally, the FEB actively connects with large domestic forest enterprises, develops industries (e.g., wood management and bamboo processing) and under-forest economies, and builds forestry bases such as Chinese fir forest, *Camellia oleifera*, and moso bamboo. It also under-forests traditional Chinese medicine and flower seedlings, carries out order management, improves the short- and long-term income of state-owned forest farms, and alleviates financial pressure.

To address financing difficulties, the FEB built a multi-faceted financial service system to effectively solve the problem of the lack of funds for tourism companies, explore the introduction of industrial and commercial capital to develop new forms of rural industry, actively develop forest health care, rural tourism, and other forestry industries, and promote the diversified development of the forestry industry. The FEB established a policy-backed bonding company—the Lvchang Forestry Financing Bonding Company—together with the Nanping Rongqiao Bonding Company to provide financial bonding services for 'Forestry Plus', whereby industrial entities and individual forest farmers can receive loans at a maximum leverage factor of 15 and at the benchmark interest rate. Moreover, the Shunchang County government encourages commercial banks to provide low-interest loans to state-owned forest farms through financial discount.

## 5. Effects

### 5.1. Organisational Framework of Government Leadership and Multi-Body Participation

In 2018, Shunchang County began considering the construction of a pilot national FEB. It looked at establishing a management, development, and operation platform focused on forestry resources based on the DICO concept and explored the creation of a model consisting of 'five inputs-outputs'. The Decentralised inputs, aimed at decentralising forest resources based on the determination and registration of rights, are: (1) to implement reforms that aim to separate ownership rights, contract rights, and usage rights of rural land; (2) to centralise resources in the FEB through transfers, leases, and trusteeships without changing the ownership of forest land; (3) to implement centralised stockpiling of diffuse and fragmented forest resources to scale up and transform the industry; (4) to integrate and utilise dispersed natural resources on a large scale; and (5) to package forest resources into centralised and contiguous high-quality and efficient assets. The centralised outputs refer to the provision of raw material bases, finance, policy, market, and tourist industry of five services through the FEB; attracting market-based funding and professional operators; taking care of investment promotion and business development; intensively managing and developing forest resources; transforming resources into assets and capital; and transforming Shunchang County's forest resource advantage into an economic advantage for local development. During the process of the capitalisation of natural resources, the value of resources can be separated from their physical medium and owned by investors in the form of natural capital, causing natural capital to move toward the highest value-added resource by market transactions. Thus, realising its value and maximising the value-added quality is possible in this manner [95].

*5.2. Leveraging Specialized Strengths to Turn Diffuse Resources into Assets*

In the FEB actor network, the integration of resources and utilisation of comparative advantages of various actors were essential for centralising fragmented forest resources. The Shunchang County government provided its county's FEB with CNY 30 million in capital. With the support of Shunchang County State-Owned Forest Farm, the Lvchang Forestry Resources Operation Company was established with two centres and three companies (Data Management Center, Asset Evaluation and Purchasing Center, Lvchang Forest Management Company, Lvchang Trusteeship Company, and Forestry Financial Services Company) under it. Lvchang Forest Management and Lvchang Trusteeship handle specialised and large-scale stockpiling, trusteeships, and management of forest resources. Forestry Financial Services provides bonding services for large businesses and professional operators to solve capital shortages and financing difficulties. The Data Management Center provides data and technical support, manages forest land under the FEB, and ensures the sound operation of the project. The Asset Evaluation and Purchasing Center is primarily responsible for pricing forest resources, providing authoritative asset evaluations, and protecting the interests of all parties. A dedicated team methodically manages and protects the forests, so that the value of the ecological capabilities and resources of forests continually increase.

*5.3. Developing Scaled-Up, Professional, and Industrial Operations*

Developing new industries and business formats within the forestry industry, such as further-processing of timber, modern agriculture, and rural tourism attracted market-based funding and professional operators. Professional operators are responsible for managing collective resources using professional mechanisms to increase the incomes of rural households and effectively utilise forest resources. By continuously expanding the FEB model, Shunchang County was able to solve the issues of the lack of rural labour, idle forest resources, low efficiency of forestry business practices, and the lack of diversity in forestry operations. The packaging of land into contiguous areas and platform operations led to the effective development of forest resources, attainment of ecological industrialisation and high-quality development of the forestry industry, and consolidation of the industrial foundation for rural revitalisation.

The centralised and integrated use of forest resources led to the scaled-up management of the forestry industry in Shunchang County. Based on the advantages offered through scaled-up management by the state-owned forest farm, the FEB was able to take control of 4385 hectares of forest land by the end of June 2020, including 871 hectares of joint-stock cooperative and leased forest land, 30.4 hectares of purchased commercial forest, and 25 hectares of forest managed under trusteeship (Data source: survey from state-owned forestry farm in Shunchang County). At the same time, to promote scaled-up forestry management, the FEB used CNY 420 million of funding from the CDB to offer 257 collateral loans totalling CNY 225 million for forest land covering 5882 hectares (Data source: survey from Fujian Lvchang Forestry Resouces Operation Co., Ltd., Nanping, China). In addition, 320 poor households in Shunchang County participated in trusteeships of forest resources, covering a forest area of 128 hectares, with each household receiving an annual income of approximately CNY 12,000 (Data source: local statistics of Shunchang County Government).

The FEB successfully implemented Fujian Province's first forestry carbon sequestration project. The first phase involved 155,500 tonnes of carbon sequestration worth CNY 2.9 million Furthermore, in June 2020, the FEB also independently planned the province's first bamboo forest carbon sequestration project, which involved 2267 hectares of land, and completed China's first bamboo forest carbon sequestration transaction, involving 69,000 tons and worth CNY 1.2 million. These initiatives have further enhanced the FEB's model for generating forestry income.

Regarding the quality of forest products, the FEB uses Lvchang Forestry Resources Operation Company to implement a wide range of contract agriculture projects, cooperate with leading forestry companies, carry out specialised and large-scale planting, meet international demand, and achieve dual certification of its forest management. A total of

18,536 hectares of forest land and 1450 hectares of bamboo forest falls within the scope of certification and provide over 40,000 m$^3$ of fir and over 30,000 culms of moso bamboo to leading companies, such as Shengsheng Wood and Yingchang Bamboo every year (Data source: survey from the state-owned forest farm in Shunchang County). This provides stable timber sales and guarantees the interests of multiple actors in the industry chain. It also supports exports to European and American markets through large-scale processing enterprises. In addition, Shengsheng Wood and others have opened poverty alleviation workshops, which have helped 50 people from poor households find jobs (Data source: survey from Fujian Shengsheng Wood Industry Co., Ltd., Sanming, China). By the end of June 2020, 137 people were employed in new poverty-alleviating public service jobs (Data source: survey from Shunchang County Government).

The FEB has borrowed from the capital market to promote the capitalisation of forest resources and has extensively sought industrial and commercial capital to develop new rural industries and business types. It has teamed up with property rights trading institutions, such as the China Beijing Equity Exchange and the Haixia Equity Exchange, to trade large-scale forest resource portfolios. The FEB also teamed up with external professional operators in tertiary industries, such as tourism development companies, to professionally manage forest resources and promote the expansion of the forestry industry into the service sector. As the end of June 2020, the FEB has invested an estimated CNY 16.7 billion in tourism projects. These include 11 pilot projects under construction (namely, projects at Xikeng, Qishan, Guanjingshan, Ganshan, Longshan, and Luying Nursery as well as on the under-forest economy) and 13 projects that have passed the initial planning stage (including a meditation town, the Citrus Culture Innovation Park, the Yangkou Red Tourism Flower Town, and the Dasheng Amusement Park) (Data source: survey from Fujian Lvchang Forestry Resources Operation Co., Ltd., Nanping, China).

## 6. Conclusions and Policy Recommendations

The analysis in this study indicates that China's FEB project achieved the capitalisation of forest resources in two ways.

First, mobilising the enthusiasm of many actors was a necessary condition for the capitalisation of forest resources. In the past, governments supported projects through subsidies. Shunchang County took a different approach with the FEB project, building an actor network by connecting diverse entities and integrating them through mechanisms that linked their interests. The Shunchang County government assumed the role of the principal actor in the network, coordinated the interests of the heterogeneous actors, and ensured their interests were closely aligned. It was vital to the construction of the FEB, as it is difficult to achieve development by relying solely on the principal actor.

Second, the scale of forest resources was central to their capitalisation. Following the tenure reform of collective forests in China, most forestlands in counties are contracted to rural households. Approximately 77% of the forest land in Shunchang County was in the hands of collectives and individuals, which presented an obstacle to the centralised large-scale utilisation of forest resources. Applying the DICO model, the FEB scaled up the management of fragmented forest resources, motivated downstream actors, such as forest farms, operators, and processing companies, and increased the incomes of upstream actors, such as collectives and farmers, in the process of industrialisation.

There are some limitations in our analysis. The FEB controls more than 4000 hectares of forest, but this accounts for only 3.29% of China's total forest land contracted to collectives and individuals. There is still scope for significant expansion. The project has achieved remarkable results by offering four transfer methods (stock ownership, trusteeship, lease, and buyout). In further research, specified mechanisms should be formulated for the varying circumstances of part-time farmers, forest farmers, poor households, and rural residents who travel to cities. However, ANT is still a useful tool for analysing the FEB project. The case of Shunchang County offers us some new insights to understand the FEB and some policy recommendations.

First, the multiple parties involved must be motivated in a coordinated manner. Special attention should be paid to the coordinating and motivational role of the principal actor. If a government is a principal actor, it must move beyond the old notion of using subsidies to promote projects and focus on leveraging policy funds. By improving mechanisms that align interests, such as dividends, profit remittance, employment opportunities, and service provision, as well as motivating diverse actors to develop the agriculture industry, all participating actors can benefit from an FEB.

Second, forestry resources must be scaled up. Except for land controlled by state-owned forest farms, most of China's forest land is scattered among village collectives and rural households. Due to China's ongoing urbanisation and industrialisation, many rural residents travel to cities for work, resulting in idle forest resources that are not used effectively. Consequently, targeted measures are needed to achieve the large-scale utilisation of China's diffuse forest resources.

Third, the actor network must be stabilised by balancing the interests of the various actors. Despite the remarkable results achieved by the FEB, weak links remain between rural households and the enterprises involved. Diverse measures are needed to provide solutions tailored to the circumstances of rural households to motivate them to participate. Various types of agricultural industrial groups should be developed to promote close cooperation between enterprises, cooperatives, and members; income channels must be expanded and diversified; and more enterprises must be included in the FEB to ensure its stability. Finally, banks, bonding companies, and other market entities should be encouraged to participate through profit-sharing to increase financial support for scaled-up operations.

**Author Contributions:** Conceptualisation, G.W. and Y.W.; methodology, G.W.; validation, X.K. and Y.W.; formal analysis, G.W. and Q.G.; investigation, G.W.; resources, G.W. and Q.G.; writing—original draft preparation, G.W. and Y.W.; writing—review and editing, Y.W.; visualisation, G.W.; supervision, X.K.; project administration, Y.W.; funding acquisition, Y.W. All authors have read and agreed to the published version of the manuscript.

**Funding:** This research was funded by the Fundamental Research Funds for the Central Universities and the Research Funds of Renmin University of China (No. 20XNA018).

**Institutional Review Board Statement:** Not applicable.

**Informed Consent Statement:** This article does not contain any studies with human participants or animals performed by any of the authors. Informed consent was obtained from all the individual participants included in the study.

**Data Availability Statement:** The author may provide raw data if necessary.

**Conflicts of Interest:** The authors declare no conflict of interest.

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
