# Peer review of "China’s Forest Eco-Bank Project: An Analysis Based on the Actor-Network Theory"

_forests, doi:10.3390/f13060944_

Round 1

Reviewer 1 Report

The article describes China’s Forest Eco-Bank (FEB) project and analyze it based on the actor-network theory. Authors identified human and non-human actors for FEB, network connections between them. Consolidation of the decentralized forest resources and mobilization of heterogenous participants is considered, including details about rental and loan mechanisms. Results achieved by FEB are summarized.

Overall, the article makes a good impression. But I think it should be published in another journal – with the Aims and Scope more in Economics and Management areas. Also, it is of interest to a limited number of scientists – due to the large number of details about the payments for merged forest lands. Too little information is given about data collection process. It’s not clear what was the composition of the interview teams, what was the ratio of people in charge to rural residents? What methods were used to analyze data? The Results section lacks a visual representation of the comparison of the initial data and the results achieved by FEB, changes in involved forest areas in the form of a table, map or diagram.

In line 505 “a monthly interest of 4.89%” – it means “annual interest” or not? In References some links are not available (error 404 or timeout) – it’s numbers 87, 88, 90, 91, 93, 94.

Author Response

Point 1: But I think it should be published in another journal – with the Aims and Scope more in Economics and Management areas. Also, it is of interest to a limited number of scientists – due to the large number of details about the payments for merged forest lands.

Response 1: Thank you for your comments. About Forests, the subject areas include “forest ecology, management, and restoration” and “forest economics, natural resource policy, and planning”. We believe that this paper will be of interest to the readership of Forests because our findings suggest that the Chinese government played an important role as the principal actor in moving the forest eco-bank (FEB) project. By mobilising various actors, ensuring that their interests were aligned, and scaling up the management of forest resources, the FEB project was able to capitalise on its forest resources. Our research presents important implications that could help China realise the economic and ecological value of its forest resources, which have been negatively affected by the high degree of fragmentation and unsustainable exploitative practices.

Point 2: Too little information is given about data collection process. It’s not clear what was the composition of the interview teams, what was the ratio of people in charge to rural residents? What methods were used to analyze data? The Results section lacks a visual representation of the comparison of the initial data and the results achieved by FEB, changes in involved forest areas in the form of a table, map or diagram.

Response 2: Thank you for your comments. We have added more details of our data resources in section 3.2 by using the tracking model of our manuscript. You can trace our changes in the updated version by choosing the “Show Markup” in the “Review” column. In our paper, we use the actor-network theory (ANT), which is a framework and methodology for a qualitative study, to analyze the text as data. We rephrase the title of the fifth section as “Effects” to distinguish our case study from general empirical studies. We also added data sources and corrected some of the data in this section to strengthen our analysis. The effects of FEB include reconstruction of actors by government leadership and multi-party participation, leveraging specialized strengths to turn diffuse resources into assets, and developing scaled-up, professional, and industrial operations.

Point 3: In line 505 “a monthly interest of 4.89%” – it means “annual interest” or not?

Response 3: Thank you for your comments. We recalculated the interest number and change it to annual interest.

Point 4: In References some links are not available (error 404 or timeout) – it’s numbers 87, 88, 90, 91, 93, 94.

Response 4: Thank you for your comments. We checked all the references and found some websites had been timeout. So we updated the links in the new version.

Reviewer 2 Report

This research paper touches upon an important issue within the debate forest management and actor role. Base on field research it proposes an analysis of cooperation among various actors 17 in the implementation process of the forest eco-bank project. This research gives a good overview on actor-based theory on forest management.

 This paper shows an innovative approach the actor-network theory framework to assess forest actors. The conceptual and the theoretical scheme are not quite clear. Authors need to provide a good analytical frameworks. The structure of this paper is acceptable.

Before providing detailed comments to the specific sections, I have some general suggestions to strengthen the analytical consistency.

All references need to be rechecked and well reframe

Overall comment

The authors need to propose conceptual and the theoretical framework. Result section can be more strengthen.

Line 146: explain concretely these two concepts

Line 277: a global map showing the site location in china is necessary

Line 301: state the base of this categorization

Table 1: what drive this classification??

Line 305: rephrase

Line 336: a reference to the table is mandatory

Line 724: what? it is now that you are coming with result??

Figure 5: something is missing?

Figure 5: is there any key for this figure?

Line 836: sometime this is an advantage

Line 842: is there and alternative model?

Line 847: how such a small impact?

All remarks and comments are in the manuscript.

Hope these comments are helpful to improve the manuscript for submission for Forest Economics, Policy, and Social Science.

Author Response

Point 1: Authors need to provide a good analytical frameworks.

Response 1: Thank you for your comments. We rephrased some sections’ titles and added more details to distinguish our case study. You can trace our changes in the updated version by choosing the “Show Markup” in the “Review” column in MS-word.

Point 2: All references need to be rechecked and well reframe.

Response 2: Thank you for your comments. We checked all the references and found some websites had been timeout. So we updated the links in the new version.

Point 3: The authors need to propose conceptual and the theoretical framework. Result section can be more strengthen.

Response 3: Thank you for your comments. You can trace our changes in the updated version by choosing the “Show Markup” in the “Review” column in MS-word.

Point 4: Line 146: explain concretely these two concepts.

Response 4: Thank you for your comments. Actor-network theory (ANT) is a theoretical and methodological approach to social theory where everything in the social and natural worlds exists in constantly shifting networks of relationships. Thus, objects, ideas, processes, and any other relevant factors are seen as just as important in creating social situations as humans. We added some descriptions of the two concepts in the updated version.

Point 5: Line 277: a global map showing the site location in china is necessary

Response 5: Thank you for your comments. We have updated the map.

Point 6: Line 301: state the base of this categorization. Table 1: what drives this classification?? Line 336: a reference to the table is mandatory

Response 6: Thank you for your comments. We followed the framework of Callon (1984) in references [74]. Honestly, the classification is some kind of subjective but it followed the essential rules of ANT.

Point 7: Line 305: rephrase

Response 7: Thank you for your comments. We rephrased some tiers titles in this section to distinguish our case study from general empirical studies.

Point 8: Line 724: what? it is now that you are coming with a result??

Response 8: Thank you for your comments. We rephrased some descriptions to strengthen our case analysis and exchanged the position of figure 3 and figure 4 to make the process clear.

Point 9: Figure 5: something is missing? Is there any key for this figure?

Response 9: Thank you for your comments. Given the context, we delete this figure to make the analysis more clear and concise. We added data sources and corrected some of the data in this section to strengthen our analysis.

Point 10: Line 836: sometimes this is an advantage. Line 842: is there and alternative model?

Line 847: how such a small impact?

Response 10: Thank you for your comments. We rephrased these sentences to make the conclusion clear.

Round 2

Reviewer 1 Report

Thanks to the authors for taking into account the comments and revision of the paper. Now your work looks better and is more suitable for publication in Forests. But I ask you to check the cited references again, some are still not opening due to a timeout server error (for example, 95, 92) - perhaps these are local China's sites, inaccessible from other countries?

Author Response

Thank you for your suggestions and comments. We checked all the links again and changed their format. Now all of them can be checked the links by pressing the "Ctrl+Click" to open the websites by Word software. Thank you.
